# Empowering Memory Assistance: An Episodic Memory-Based Framework for Personalized Recommendations

## Abstract

Artificial agents operating in dynamic environments require the ability to recall and contextualize past experiences to inform future behavior. Drawing inspiration from human episodic memory, we propose a cognitively grounded recommendation framework that models time-evolving personal experiences using a dynamic, multimodal memory architecture. Our system encodes temporally structured actions, places, and interactions into a hierarchical temporal graph network (TGN), enabling agents to disambiguate overlapping behavior patterns and anticipate future actions based on long-term experience. Unlike traditional recommendation or forecasting models that rely on static, task-specific patterns, our approach supports continual memory updates without retraining, and generalizes across varied activity sequences. Evaluated on a structured dataset derived from three years of egocentric recordings, our model significantly outperforms state-of-the-art baselines (e.g., AntGPT, DyRep, Palm) on next-activity prediction and sequence alignment metrics. This work introduces a scalable, cognitively inspired memory architecture with broad applications in lifelong learning, assistive robotics, and human-AI collaboration.

## 1 Introduction

Episodic memory allows humans to recall experiences richly contextualized by time, place, and activity Tulving (2002), supporting planning, anticipation, and adaptive decision-making. Artificial agents operating in dynamic human environments require a similar capability: they must accumulate experience continuously, track long-term behavioral patterns, and adapt to shifting contexts without retraining. This requirement is especially critical in assistive settings—for example, a personalized robotic assistant supporting users who may have difficulty managing or recalling daily activities, where stable long-term recall and reliable next-activity anticipation are essential for safe and adaptive interaction.

Recent foundation models such as PaLM (Chowdhery, Narang, and Devlin, 2022) and AntGPT (Zhao, Wang, Zhang, Fu, Do, Agarwal, Lee, and Sun, 2024) exhibit strong reasoning abilities but lack persistent memory, continuity of experience, and sensitivity to the fine-grained temporal–semantic structure underlying human behavior. Their reliance on static language inputs limits their ability to model routines, multimodal signals, and evolving real-world contexts.

We address this gap by introducing a cognitively inspired episodic-memory framework built on Temporal Graph Networks (TGNs) (Rossi, Chamberlain, Frasca, Eynard, Monti, and Bronstein, 2020). Unlike prior uses of TGNs—typically applied to communication logs or transactional streams—our approach treats the TGN as a *memory substrate* in which each event becomes a multimodal node encoding time, activity, and place. These nodes evolve as episodic memories enriched with visual, speech, and textual cues, enabling the system to distinguish visually similar actions performed at different times or in different locations.

As illustrated in Fig. 1, the agent receives either fine-grained micro-actions (e.g., cutting, boiling) or higher-level activities (e.g., cooking, walking). Each observation is stored as a *time–activity–place* memory event, allowing the agent to recommend both detailed next steps and higher-level activity transitions depending on context.

Human activity is inherently temporal and contextual, spanning micro-actions, hour-scale activities, daily routines, and weekly or seasonal cycles. Standard TGNs timestamp events numerically but overlook this multi-scale structure. We therefore integrate wavelet time encoding, calen-

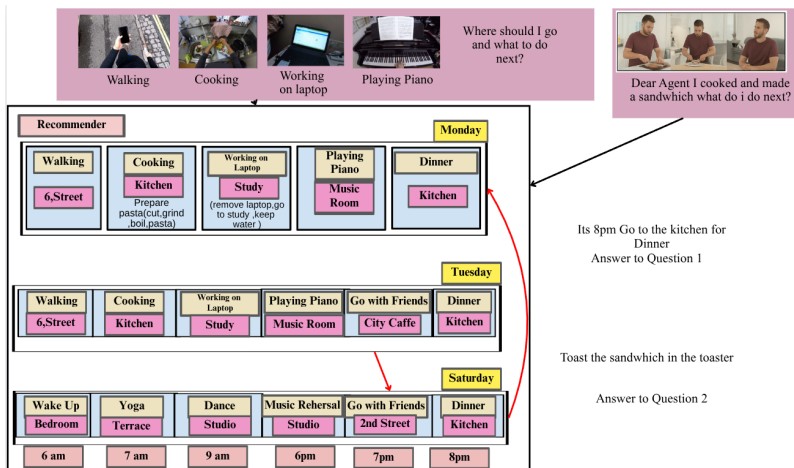

Figure 1: The agent's episodic-memory representation. Fine-grained actions or activity-level events are stored as time–activity–place nodes (yellow = action, pink = place), enabling next-step recommendations at multiple levels of abstraction.

dar embeddings, and day-wise episodic segmentation to capture patterns ranging from minutes to years. By jointly modeling interactions among time, activity, and place—rather than fusing them independently—the system learns meaningful co-occurrence patterns (e.g., "evening cooking except on weekends") that simple fusion baselines cannot capture.

These design choices—multimodal fusion, temporal attention, and an evolving TGN memory—enable long-horizon reasoning, anticipatory prediction, and personalized assistance. In contrast to baselines that encode time, activity, and place separately and combine them via rank fusion, our architecture learns their interactions through experience, discovering micro-action sequences and context-specific regularities.

Recent LLM agents (e.g., ChatGPT, Gemini) include mechanisms labeled as memory, but these rely on short-lived context windows (Brown et al., 2020) or static document-retrieval databases (Lewis et al., 2021). Neither provides an episodic representation grounded in temporally ordered, multimodal what–where–when events. They cannot track evolving routines or perform long-horizon temporal prediction across days or contexts. Our framework addresses this gap by maintaining a persistent multimodal episodic structure that updates with experience and supports next-activity prediction.

Existing egocentric datasets like EPIC-KITCHENS-55 (Damen et al., 2020) and EGTEA Gaze+ (Li et al., 2020) focus on short-horizon, single-environment tasks such as cooking, and therefore lack the episodic boundaries and multi-domain activity transitions required for next-activity prediction. Hence we rely on the broader Ego4D dataset, which we reorganize into structured day-level timelines to capture real-world episodic boundaries and cross-domain behavioral transitions. This organization enables evaluation aligned with our scenario: personalized activity and action anticipation for assistive robots that support users who may struggle to manage or recall daily activities.

**Contributions.**

- A multimodal episodic-memory TGN architecture that reorganizes nodes into *time–activity–place* memory events with learnable cross-modal fusion.

- A day-wise episodic memory structure with wavelet- and calendar-aware temporal encoding capturing both micro-action chains and long-horizon periodic patterns.

- State-of-the-art performance on Ego4D (Grauman et al., 2022; Zhou et al., 2025) for short- and long-horizon forecasting.

- Evidence that the model becomes increasingly sensitive to *time–activity–place* interactions and co-occurrence statistics.

- Demonstrations of anticipatory prediction, personalized assistance.

Throughout the paper, we use the term "user" to refer to the human whose behaviors and contexts the agent continuously observes and supports.

## 2 RELATED WORK

**Graphs for Episodic Memory Recommendation.** Static graphs fail to capture temporal dynamics, evolving relationships, and incremental node additions, limiting their utility for episodic memory recommendation. Graph Neural Networks (GNNs) (Jin, Song, and Shi, 2019; Wang, Jiang, Syed, Conway, Juneja, Subramanian, and Chawla, 2020; Song, Li, Chang, Xie, Hao, and Qin, 2024) perform well in node classification but struggle in sequential, evolving contexts such as life-logging. Dynamic graph learning addresses this by modeling temporal interactions, with applications in social networks, transportation, and biology (Barros, Mendonça, Vieira, and Ziviani, 2021; Skarding, Gabrys, and Musial, 2021; Xue, Zhong, Li, Chen, Zhai, and Kong, 2022; Kazemi, Goel, Jain, Kobyzev, Sethi, Forsyth, and Poupart, 2020; Kumar, Zhang, and Leskovec, 2019). Benchmarks such as DGB, TGB, and TransactionTempGraph (Poursafaei and Huang, 2022; Huang and Poursafaei, 2024; Zhang, Luo, Lu, and He, 2024) and toolkits like DyGLib (Trivedi, Farajtabar, Biswal, and Zha, 2019; Yu, Sun, Du, and Lv, 2023b) provide temporal data resources, but often rely on limited features (e.g., bag-of-words, word2vec (Katz, 1985; Mikolov, Chen, Corrado, and Dean, 2013)) and lack support for ordered sequences or contextual reasoning. We extend temporal graph learning by incorporating spatial, temporal, and sequential cues for personalized recommendations, and by leveraging Text-Attributed Graphs (TAGs) (Sen, Namata, Bilgic, and Getoor, 2008; Wang and Shen, 2020; Yang, Liu, Xiao, Li, Lian, and Agrawal, 2021; Yan, Li, Long, Yan, and Zhao, 2023) enriched with large language models (Yu, Ren, Gong, Tan, Li, and Zhang, 2023a; He, Bresson, Laurent, Perold, LeCun, and Hooi, 2023; Pan, Zhang, Zhang, Hu, and Zhao, 2024; Tang, Yang, Wei, Shi, and Su, 2024; Ye, Zhang, Wang, Xu, and Zhang, 2024; Zhao, Zhuo, Shen, Qu, and Liu, 2023), bridging temporal, contextual, and visual modalities.

**Action Anticipation.** Classical models (Kingma, Salimans, Jozefowicz, Chen, Sutskever, and Welling, 2016; Kingma and Dhariwal, 2018; Rezende and Mohamed, 2015) predict actions from sequence patterns but lack long-term memory and personalization. Marked Temporal Point Processes (MTPPs) (Hawkes, 1971; Du, Dai, Trivedi, Upadhyay, Gomez-Rodriguez, and Song, 2016; Mei and Eisner, 2017; Zhang, Lipani, Kirnap, and Yilmaz, 2020; Zuo, Jiang, Li, Zhao, and Zha, 2020) model continuous-time events, while recent flow-based methods (Shchur, Biloš, and Günnemann, 2020; Mehrasa, Deng, Ahmed, and Chang, 2019) improve sampling efficiency but remain limited in multi-context generalization. Normalizing flows (Rezende and Mohamed, 2015; Mehrasa, Deng, Ahmed, and Chang, 2019) provide tractable sampling yet fail to adapt to evolving, personalized behaviors. Recent state-of-the-art approaches—Palm, AntGPT (Zhao, Wang, Zhang, Fu, Do, Agarwal, Lee, and Sun, 2024), iCVAE (Mascaro, Ahn, and Lee, 2024), ObjectPrompt (Zhang, Fu, Wang, Agarwal, Lee, Choi, and Sun, 2023), and Replai Mittal et al. (2022b)—leverage large-scale datasets such as Ego4D (Grauman, Westbury, Byrne, and Chavis, 2022) and EPIC-Kitchens Damen et al. (2018) but require retraining for new tasks and cannot model episodic memory or personalized anticipation. Our approach addresses these gaps by unifying temporal graphs, dynamic memory representations, and log-normal flows to enable multi-activity, long-term anticipation and personalized recommendations in evolving environments.

## 3 APPROACH

In our setting, the agent continuously observes its human companion (the "master") performing everyday activities across varied temporal routines (e.g., structured weekday mornings versus flexible weekends). Each observation is encoded as an episodic tuple (character, action, time, place), where the master is the central character and time is recorded with fine granularity.

Actions denote fine-grained primitives (e.g., *cut*, *stir*, *pour*), while activities compose these into higher-level routines (e.g., *cooking*). The agent fuses visual activity embeddings with action tokens extracted from spoken commands using Whisper (Radford et al., 2022), enabling robust recall even under occlusion or missing visual cues. These multimodal episodes accumulate in structured memory modules that support pattern discovery and improved action recommendation.

Our model performs future action sequence forecasting: given multimodal episodic memory, it predicts a variable-length sequence of future actions. We optimize this output using Connectionist Temporal Classification (CTC), which aligns predicted and ground-truth action tokens without requiring pre-segmented annotations, making the task a natural sequence forecasting problem. The figure 2 illustrates the full pipeline of our episodic-memory recommender, showing how multimodal events are encoded, stored in a temporal graph, updated over time, and used to predict future actions.

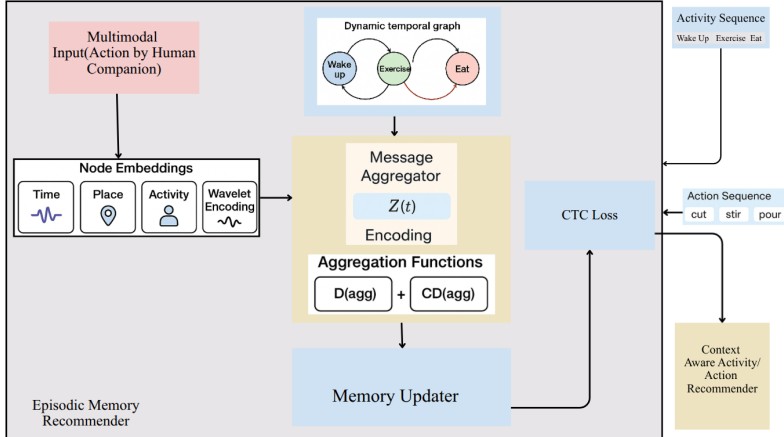

Figure 2: Overview of the episodic-memory recommender. Multimodal events (vision, speech, time, place, activity) are encoded as time–activity–place nodes in a dynamic temporal graph. Daily and cross-day message aggregation updates memory states, while temporal attention and wavelet-based time encoding capture both micro-action chains and long-horizon routines. The final episodic memory supports next-action and sequence-level forecasting.

## 3.1 EPISODIC MEMORY RECOMMENDER

Building on the agent's multimodal episodic encoding of the user's daily and weekly routines, the episodic memory recommender maintains a structured, continuously evolving lifelog. Each observed event—defined by its *time*, *activity*, and *place*—is stored as a node in a dynamic temporal graph. As the agent observes activities (e.g., *cooking* composed of micro-actions such as *cut* or *stir*) across different contexts, the graph accumulates regularities and variations in behavior.

To model these temporal dynamics, we employ a Temporal Graph Network (TGN) in an encoder–decoder architecture. The encoder processes the continuous-time event stream and updates node representations based on timestamps and interactions. At time $t$, the set of node embeddings is:

$$\mathbf{Z}(t) = \big(\mathbf{z}_1(t), \ldots, \mathbf{z}_{n(t)}(t)\big), \tag{1}$$

where $\mathbf{z}_i(t)$ is the embedding of node $i$ and $n(t)$ is the number of events observed so far. Each embedding integrates multimodal cues and temporal context, enabling long-horizon activity anticipation.

**Node Embeddings.** At each time step $t$, the agent collects available multimodal signals and creates an event node when it detects an activity or receives activity-related speech.

*Visual stream.* Visual observations provide object- and action-level cues, capturing micro-actions (e.g., *cut*, *stir*, *pour*) underlying higher-level activities.

*Speech and language stream.* If speech is present (e.g., "add salt here"), the utterance is transcribed and encoded, supplying complementary information about goals or intent that may be absent in the visual stream.

*Multimodal fusion.* Visual and linguistic features are combined with time, activity, and place encodings to form a unified event representation. For an event at time $t$, the multimodal node embedding is:

$$\mathbf{z}_i(t) = f(\mathbf{e}_{\text{time}}, \mathbf{e}_{\text{activity}}, \mathbf{e}_{\text{place}}, \mathbf{e}_{\text{vision}}, \mathbf{e}_{\text{text}}), \tag{2}$$

where $\mathbf{e}_{\text{vision}}$ and $\mathbf{e}_{\text{text}}$ are the extracted visual and linguistic embeddings. The system naturally handles missing modalities by falling back to the available encoders, ensuring consistent construction of event nodes within the temporal graph.

*Modality flexibility.* Since the agent functions in a dynamic, real-world environment, not all modalities are available at all times. When only one modality is present (e.g., silent visual observation or text-only interaction), the system automatically falls back to the corresponding encoder. This design preserves a consistent interface for constructing node representations, enabling uniform downstream processing within the temporal graph.

**Daywise Memory Representation.** Building on the multimodal node embeddings described above, the agent organizes its episodic memory in a day-structured form that mirrors natural human recall. Each node embedding $\mathbf{z}_i(t)$ reflects an event the agent observed, and these events accumulate

chronologically throughout the day. This yields daily memory traces such as:

$$\text{DAY 1: } \{6\text{:}00 \text{ AM wake up, } 6\text{:}30 \text{ AM exercise, } 7\text{:}15 \text{ AM cooking}, \dots\},$$

where higher-level activities (e.g., *cooking*) implicitly contain the fine-grained micro-actions captured in the visual and linguistic streams (e.g., *retrieve pan*, *chop onions*, *add oil*).

Formally, when a new event occurs at time $T_j$, we identify prior events from the same day:

$$\mathcal{N}_{\text{day}} = \{i \mid T_i.\text{date} = T_j.\text{date} \wedge T_i.\text{time} \leq T_j.\text{time}\}, \tag{3}$$

where $T_i.\text{date}$ and $T_i.\text{time}$ denote the calendar date and within-day timestamp of event $i$. If $\mathcal{N}_{\text{day}} \neq \emptyset$, the existing memory for that day is refined by updating each node $i \in \mathcal{N}_{\text{day}}$:

$$\mathbf{z}_i(t+1) = f_{\text{update}}\Big(\mathbf{z}_i(t), \, \mathbf{e}_{\text{activity}_j}, \, \mathbf{e}_{\text{time}_j}, \, \mathbf{e}_{\text{place}_j}\Big), \tag{4}$$

allowing the system to accumulate and contextualize routines as the day progresses. If $\mathcal{N}_{\text{day}} = \emptyset$, the event marks the start of a new day, and a new memory node is created:

$$\mathbf{z}_j(t+1) = f\Big(\mathbf{e}_{\text{time}_j}, \, \mathbf{e}_{\text{activity}_j}, \, \mathbf{e}_{\text{place}_j}\Big). \tag{5}$$

This daywise structure emerges organically as the temporal graph evolves, producing coherent daily episodes that integrate both high-level activities and their underlying micro-actions.

**Final Memory Representation.**   At the end of each day, the system consolidates the events stored in the temporal graph into a single daily memory embedding. This representation summarizes the day's high-level activities together with the fine-grained micro-actions captured during multimodal observation, forming a compact episodic snapshot of when events occurred, where they took place, and how they unfolded.

Formally, the final memory for day $i$ is computed by aggregating the embeddings of all events associated with that day:

$$\mathbf{z}_i(t) = f_{\text{agg}}\big(\{\mathbf{e}_{\text{time}_k}, \, \mathbf{e}_{\text{activity}_k}, \, \mathbf{e}_{\text{place}_k} \mid k \in \mathcal{N}_{\text{day}}\}\big), \tag{6}$$

where $f_{\text{agg}}$ is a learnable aggregation function.

The aggregator can preserve temporal order (e.g., *wake $\rightarrow$ exercise $\rightarrow$ cook*), combine multimodal cues, or compute routine statistics such as frequency or duration. Because earlier steps encode micro-actions (e.g., *retrieve pan*, *chop onions*, *fill water bottle*), the resulting daily embedding captures both coarse daily structure and its underlying procedural details, providing a rich episodic representation for downstream anticipation and recommendation.

**Activity-Based Message Function.**   Given the daily event structure and node embeddings described above, the system updates each node's episodic state using activity-based message functions. When a new activity involving node $i$ occurs at time $t$, the agent produces a message that encodes how this observation should modify the node's memory. For an interaction between two nodes $i$ and $j$, represented by an activity embedding $\mathbf{e}_{ij}(t)$, the model generates directed messages:

$$\mathbf{m}_i(t) = \text{msg}_{\text{s}}\big(\mathbf{s}_i(t^-), \, \mathbf{s}_j(t^-), \, \Delta t, \, \mathbf{e}_{\text{action}_i}\big), \tag{7}$$

$$\mathbf{m}_j(t) = \text{msg}_{\text{d}}\big(\mathbf{s}_j(t^-), \, \mathbf{s}_i(t^-), \, \Delta t, \, \mathbf{e}_{\text{action}_j}\big), \tag{8}$$

where $\mathbf{s}_i(t^-)$ is the node's state immediately before $t$, and $\Delta t$ is the time since its last update.

If an event involves only a single node, the update is simplified to:

$$\mathbf{m}_i(t) = \text{msg}_{\text{n}}\big(\mathbf{s}_i(t^-), \, t, \, \mathbf{e}_{\text{action}_i}\big). \tag{9}$$

The learnable functions $\text{msg}_{\text{s}}$, $\text{msg}_{\text{d}}$, and $\text{msg}_{\text{n}}$ determine how new observations—including the fine-grained micro-actions embedded earlier—are incorporated into the evolving episodic memory, ensuring consistent integration across the day's event stream.

**Message Aggregator (per-day vs. cross-day influence).**   Following the message generation step, the system aggregates all messages associated with a node $i$. When multiple activities involve the same node within a batch, their messages are combined as:

$$\bar{\mathbf{m}}_i(t) = \text{agg}(\mathbf{m}_i(t_1), \, \dots, \, \mathbf{m}_i(t_b)), \tag{10}$$

where $\text{agg}$ applies frequency-aware weighting and a temporal decay kernel. This connects directly to the daywise memory structure established earlier.

*Per-day aggregation* groups messages originating from the same calendar day, strengthening the local context of that day's episodic node (e.g., an ordered morning routine).

**Cross-day influence** applies a decay function $\kappa(\Delta\text{days})$ to messages from different days. This allows recurring patterns (e.g., a weekend jog) to reinforce each other across days, while isolated anomalies gradually diminish.

Together, these mechanisms enable the model to capture both within-day event structure and multi-day periodicities by appropriately weighting same-day and cross-day messages.

**Memory Updater.** After aggregation, the updated message $\bar{\mathbf{m}}_i(t)$ is integrated into the node's episodic state through:

$$\mathbf{s}_i(t) = \text{mem}\big(\bar{\mathbf{m}}_i(t),\, \mathbf{s}_i(t^-)\big), \tag{11}$$

where $\text{mem}$ is implemented as an LSTM. This module connects directly to the per-day and cross-day aggregation scheme by determining how strongly new information influences the long-term memory of node $i$.

The LSTM maintains a gated internal state that balances *long-term regularities* (e.g., morning routines, weekday vs. weekend patterns) with *short-term variability* (one-off events or context-specific behaviors). Frequently occurring micro-actions within a day (e.g., *retrieve pan → chop vegetables → stir*) reinforce stable procedural chains, while infrequent actions naturally fade unless repeated across days. Because messages are already conditioned by temporal encodings and the per-day vs. cross-day weighting, the LSTM captures both fine-grained action sequences and broader multi-day periodicities. This yields a robust episodic memory state that supports reliable activity anticipation and personalized recommendation.

**Embedding.** Building on the updated memory states, each event node retains the multimodal embedding from Eq. 2. To make its structure explicit, we highlight the core semantic components—place, activity, and time—that anchor the fused representation. These elements form the backbone of the node's embedding and are combined through the fusion function $f(\cdot)$. This structure allows the model to distinguish fine-grained variations (e.g., "cutting vegetables at 7:00 AM" vs. "cutting vegetables at 7:00 PM") even when visual cues alone may be ambiguous. During aggregation, frequency-aware weighting reinforces recurrent behaviors while temporal decay down-weights infrequent or outdated events. The resulting episodic memory at time $t$ is:

$$\mathbf{Z}(t) = \{\, \mathbf{z}_i(t) \mid i \in \mathcal{N}(t) \,\}, \tag{12}$$

providing the rich, temporally grounded memory representation used by downstream anticipation and recommendation modules.

**Time Encoding.** To support the temporally grounded memory described above, we encode each timestamp using Wavelet Time Encoding (WT) (Sasal, Chakraborty, and Hadid, 2022), which provides multi-resolution representations of temporal structure. For timestamps $t_1, \ldots, t_n$, WT yields:

$$\mathbf{z}_i(t) = \text{WT}(t_i, \text{wavelet}), \qquad i = 1, \ldots, n, \tag{13}$$

capturing both fine-grained and long-range temporal variations. Because daily activities follow human-organized periodicities, we augment WT with calendar-based embeddings:

$$\mathbf{z}_i(t) = [\mathbf{t}_i \,\|\, \delta_i \,\|\, \mu_i \,\|\, \nu_i \,\|\, \tau_i], \tag{14}$$

where $\delta_i, \mu_i, \nu_i, \tau_i$ respectively encode day, month, year, and hour. This fusion allows the model to distinguish, for example, weekday mornings from weekend evenings, reinforcing the long-horizon temporal regularities that the memory updater must track."'

**Edge Features.** Edges link event nodes across time and space, enabling the model to contextualize how activities unfold. Each edge encodes (i) the temporal gap between events and (ii) their spatial relationship:

$$\delta_{ij}(t) = \phi(t - t_j) \,\|\, \mathbf{e}_{\text{place}}(i, j), \tag{15}$$

where $\phi(t - t_j)$ represents temporal separation and $\mathbf{e}_{\text{place}}(i, j)$ captures spatial proximity or transitions (e.g., "kitchen → hallway"). These features allow message passing to reflect both short-range procedural continuity and longer-range temporal structure, complementing the time and embedding modules described above.

**Temporal Attention Layer.** At each layer $l$, node $i$ attends to itself and its temporal neighbors, each providing its representation, temporal offset, and interaction features:

$$\{\mathbf{h}_k^{(l-1)}(t)\}, \qquad \{t - t_k\}, \qquad \{\mathbf{e}_{ik}(t_k)\}. \tag{16}$$

These are fused into a context matrix:

$$\mathbf{C}^{(l)}(t) = \left[ \mathbf{h}_k^{(l-1)}(t) \,\|\, \mathbf{e}_{ik}(t_k) \,\|\, \phi(t - t_k) \right]_{k=1}^{N}. \tag{17}$$

Multi-head attention is then applied:

$$\tilde{\mathbf{h}}_i^{(l)}(t) = \text{MultiHeadAttention}^{(l)}\left( \mathbf{q}^{(l)}(t), \, \mathbf{K}^{(l)}(t), \, \mathbf{V}^{(l)}(t) \right), \tag{18}$$

and the updated node representation becomes:

$$\mathbf{h}_j^{(l)}(t) = \text{MLP}^{(l)}\left( \mathbf{h}_i^{(l-1)}(t) \,\|\, \tilde{\mathbf{h}}_i^{(l)}(t) \right). \tag{19}$$

By conditioning attention weights on temporal offsets and interaction features, this layer links naturally to the edge and time encodings above, enabling the model to integrate fine-grained micro-action chains with broader cross-day regularities.

**Sequence-Level Prediction with CTC Loss.** To anticipate future action sequences, we employ Connectionist Temporal Classification (CTC), which handles variable-length outputs and irregular temporal spacing. Given node representations $\mathbf{Z}$, the CTC objective is:

$$\mathcal{L}_{\text{CTC}} = -\log \sum_{\pi \in \mathcal{B}^{-1}(y)} P(\pi \mid \mathbf{Z}), \tag{20}$$

where $\mathcal{B}^{-1}(y)$ enumerates all valid alignments collapsing to the target sequence $y$. CTC naturally accommodates repeated or stretched micro-actions—e.g., {cut, cut, stir, boil} aligning to {cut, stir, boil}—linking this prediction layer to the temporally irregular patterns captured by the memory and attention modules. As a result, the model produces stable sequence forecasts even under noisy, partially observed, or time-dilated real-world behaviors.Illustrative example of the approach is given in B

## 4 EXPERIMENTAL SETUP

**Dataset.** We construct a structured long-horizon dataset from Ego4D (Grauman et al., 2022), which contains isolated egocentric clips but no multi-day temporal continuity. To support long-horizon episodic-memory modeling, we reorganize these clips into a synthetic three-year timeline while preserving all original Ego4D annotations: clips are not trimmed, stitched, internally reordered, or relabeled, and all scenario- and moment-level boundaries remain intact. Timeline construction follows a timetable-style sampling strategy (weekday vs. weekend templates) that assigns clips to days based on scenario categories and places them in day-level chronological order without generating intra-clip continuity. To ensure realism and statistical robustness we allow limited, tracked repetitions of clips only when necessary to satisfy day-length sampling constraints; such repetitions are logged in the dataset metadata and are applied in a split-aware manner to prevent cross-split leakage. Mild scenario-level balancing is applied at sampling to avoid dominance by highly frequent categories. The synthetic timeline is used only for temporal-graph and link-prediction experiments (Tables 2–7); all long-term anticipation (LTA) results in Table 1 are trained and evaluated exclusively on the original Ego4D LTA splits, ensuring LTA performance is not influenced by the timeline construction.Details of dataset construction are as in Appendix A.

**Implementation Details.** The model applies a time-scaling factor ($1 \times 10^{-6}$) to weight sampling by temporal intervals, where larger values prioritize recent events and 0.0 yields uniform sampling. The walk encoder employs 10 attention heads with 5-step random walks to capture temporal-structural dependencies, optimized for two NVIDIA RTX 6000 GPUs (24GB each). Temporal evolution is modeled with a fixed gap of 10 units, and embedding dimensions are set to 100 (time) and 172 (position). EdgeBank memory provides unbounded storage with optional time-window and repeat-threshold modes. Training uses Adam (lr = 0.0001), dropout = 0.1, and early stopping with patience = 5. Data is split 70%/15%/15% for train/validation/test. Multimodal features (text and image embeddings from *videorecap* (Islam, Ho, Yang, Nagarajan, Torresani, and Bertasius, 2024)) enrich node representations. We compare against DyRep with time-interval–aware neighbor sampling, and note that hyperparameters are tunable for other datasets.

## 5 EXPERIMENTS AND RESULTS

### 5.1 COMPARISON WITH LONG-TERM ANTICIPATION MODELS

In this experiment, we compare our approach against state-of-the-art long-term anticipation models on their ability to predict future activities. Table 1 compares several strong baselines, including

RUSTLM (Mittal, Morgado, Jain, and Gupta, 2022a), ICVAE (Mascaro, Ahn, and Lee, 2024), CLIP+Transformer (Radford, Kim, Hallacy, and Ramesh, 2021; Vaswani, Shazeer, Parmar, and Uszkoreit, 2023), Object Prompt (Zhang, Fu, Wang, Agarwal, Lee, Choi, and Sun, 2023), Palm, and AntGPT (Zhao, Wang, Zhang, Fu, Do, Agarwal, Lee, and Sun, 2024). These models represent current state-of-the-art methods in action anticipation.

**Metric:**

Following the Ego4D LTA benchmark proto-col (Ishibashi, Ono, Kugo, and Sato, 2023), we report Edit Distance (ED) (Przybocki, Sanders, and Le, 2006), which measures similarity between predicted and ground-truth action sequences. Lower ED indicates better predictive alignment. All evaluations were conducted on the Ego4D LTA dataset.

Overall, our episodic memory recommender achieves substantially lower Action error than all baselines, demonstrating superior ability to model long-term activity evolution. This is essential for

Table 1: Comparison of methods on the Action metric (lower is better).

| Method | Action ↓ |
|---|---|
| RUSTLM | 0.9432 |
| ICVAE | 0.9304 |
| CLIP+Transformer | 0.9290 |
| Object Prompt | 0.9276 |
| Palm | 0.9120 |
| AntGPT | 0.8853 |
| **Ours** | **0.7220** |

applications such as social companions for individuals with memory disorders, where predictions must reflect personalized routines and long-term temporal dependencies.

## 5.2 Comparison with Other Graph Models

We evaluate state-of-the-art graph-based models—widely applied in temporal domains such as social networks, transportation, and biology—on lifelog data. The task involves predicting the next activity (*Test Score*) and integrating unseen activities into an episodic memory recommender (*New Node Score*), thereby testing models' ability to support long-term anticipation and personalized activity prediction.

Table 2: Comparison of graph models on link prediction for lifelog data (mean ± std).

| Graph Model | Test ROC Accuracy | Test Precision | New Node ROC Accuracy | New Node Precision |
|---|---|---|---|---|
| JODIE | $0.7088 \pm 0.0123$ | $0.6938 \pm 0.0141$ | $0.6302 \pm 0.0187$ | $0.5936 \pm 0.0204$ |
| DyRep | $0.7792 \pm 0.0098$ | $0.7575 \pm 0.0129$ | $0.7297 \pm 0.0154$ | $0.7308 \pm 0.0171$ |
| TGAT | $0.6670 \pm 0.0137$ | $0.6436 \pm 0.0150$ | $0.5430 \pm 0.0199$ | $0.5660 \pm 0.0220$ |
| TGN | $0.6820 \pm 0.0105$ | $0.7398 \pm 0.0134$ | $0.6298 \pm 0.0176$ | $0.5840 \pm 0.0193$ |
| CAWN | $0.6776 \pm 0.0117$ | $0.7024 \pm 0.0148$ | $0.6008 \pm 0.0215$ | $0.5602 \pm 0.0208$ |
| TCL | $0.6260 \pm 0.0141$ | $0.6623 \pm 0.0160$ | $0.5505 \pm 0.0223$ | $0.5985 \pm 0.0214$ |
| GraphMixer | $0.6153 \pm 0.0129$ | $0.6550 \pm 0.0144$ | $0.5610 \pm 0.0191$ | $0.5471 \pm 0.0185$ |
| DyGFormer | $0.6791 \pm 0.0109$ | $0.6626 \pm 0.0138$ | $0.5860 \pm 0.0180$ | $0.6290 \pm 0.0172$ |
| **Ours** | $\mathbf{0.8664 \pm 0.0061}$ | $\mathbf{0.8185 \pm 0.0079}$ | $\mathbf{0.7962 \pm 0.0094}$ | $\mathbf{0.8020 \pm 0.0102}$ |

As shown in Table 2, our model outperforms all baselines across both test and new-node settings. This indicates that capturing only recent activity patterns is insufficient for lifelog recommendation. Instead, maintaining structured sequential memory at daily resolution is essential for personalized, context-aware predictions, going beyond local neighborhood or similarity-based reasoning.

## 5.3 Ablation Studies

In the ablation studies, we aim to validate our approach by changing hyperparameters and evaluating specific parameters of the model. To achieve this, we measure the Test ROC AUC score, which assesses how effectively the agent can recommend the next activity. Additionally, we analyze the New Node score to evaluate how accurately the model can incorporate new nodes into its memory. These metrics provide insights into the model's recommendation capability and its ability to adapt and expand its episodic memory effectively.

**Time Encoding.** Table 3 reports the performance of different temporal encoding methods, including Sinusoidal (Sun, Yuan, Xu, Mai, Siddharth, Chen, and Marina, 2024), Cosine (Vaswani, Shazeer, Parmar, and Uszkoreit, 2023), Wavelet (Sasal, Chakraborty, and Hadid, 2022), Fourier, Gaussian (Ren, Wang, Jia, Laili, and Zhang, 2023), and learnable matrices. Among these, Wavelet encoding achieves the best results across all metrics, highlighting its effectiveness in activity recommendation tasks. Wavelets are particularly effective as they decompose signals into both time and frequency components, enabling detection of patterns across multiple scales. This is well suited for lifelog data, where activities often follow periodic rhythms such as daily routines and seasonal variations.

Table 3: Comparison of time encoding methods for link prediction tasks (mean ± std).

| Time Encoding | Test ROC Accuracy | Test Precision | New Node ROC Accuracy | New Node Precision |
|---|---|---|---|---|
| Sinusoidal | $0.779 \pm 0.010$ | $0.705 \pm 0.012$ | $0.611 \pm 0.015$ | $0.591 \pm 0.017$ |
| Cosine | $0.758 \pm 0.011$ | $0.716 \pm 0.013$ | $0.593 \pm 0.018$ | $0.550 \pm 0.019$ |
| **Wavelet** | **$0.8664 \pm 0.0061$** | **$0.8185 \pm 0.0079$** | **$0.7962 \pm 0.0094$** | **$0.8020 \pm 0.0102$** |
| Fourier | $0.659 \pm 0.014$ | $0.547 \pm 0.016$ | $0.616 \pm 0.019$ | $0.661 \pm 0.018$ |
| Gaussian | $0.617 \pm 0.013$ | $0.537 \pm 0.015$ | $0.581 \pm 0.021$ | $0.555 \pm 0.020$ |
| Learnable | $0.736 \pm 0.012$ | $0.733 \pm 0.014$ | $0.485 \pm 0.022$ | $0.506 \pm 0.021$ |

Unlike sinusoidal or cosine encodings with fixed frequencies, wavelets provide adaptive representations that better align with natural temporal evolution. This adaptability enhances modeling of both local and global temporal dependencies, leading to consistent gains in ROC and precision metrics, and confirming the suitability of wavelet encoding for personalized activity prediction.

**Backbone Architecture Comparison.** To evaluate the effect of backbone choice on pattern-oriented link prediction, we substituted different architectures into our framework and measured performance across four metrics. Results (Table 4) show substantial variation: while models such as TGN and TGAT perform competitively, DyRep consistently achieves the best results across Test ROC Accuracy, Test Precision, New Node ROC Accuracy, and New Node Precision. DyRep's

Table 4: Backbone comparison on link prediction. DyRep (ours) outperforms all alternatives across test and new-node settings.

| Backbone | Test ROC Accuracy | Test Precision | New Node ROC Accuracy | New Node Precision |
|---|---|---|---|---|
| JODIE | $0.6118 \pm 0.0121$ | $0.6226 \pm 0.0140$ | $0.5586 \pm 0.0205$ | $0.5682 \pm 0.0189$ |
| DyFormer | $0.6010 \pm 0.0133$ | $0.6187 \pm 0.0161$ | $0.6545 \pm 0.0187$ | $0.6656 \pm 0.0174$ |
| TGAT | $0.6618 \pm 0.0118$ | $0.7023 \pm 0.0148$ | $0.6125 \pm 0.0199$ | $0.6414 \pm 0.0203$ |
| TGN | $0.7071 \pm 0.0104$ | $0.7205 \pm 0.0137$ | $0.6102 \pm 0.0181$ | $0.5779 \pm 0.0190$ |
| CAWN | $0.5688 \pm 0.0142$ | $0.5339 \pm 0.0175$ | $0.4899 \pm 0.0221$ | $0.4689 \pm 0.0218$ |
| TCL | $0.6780 \pm 0.0127$ | $0.6584 \pm 0.0150$ | $0.5921 \pm 0.0204$ | $0.5956 \pm 0.0187$ |
| GraphMixer | $0.6249 \pm 0.0131$ | $0.6386 \pm 0.0143$ | $0.5236 \pm 0.0210$ | $0.4892 \pm 0.0201$ |
| **DyRep (Ours)** | **$0.8664 \pm 0.0061$** | **$0.8185 \pm 0.0079$** | **$0.7962 \pm 0.0094$** | **$0.8050 \pm 0.0102$** |

advantage stems from its ability to capture both global and local temporal dependencies. Its event-driven formulation models evolving relationships and irregular dynamics, enabling robust performance in life-log data with mixed periodic and non-periodic behaviors. This adaptability makes it particularly effective for long-horizon sequential prediction tasks. More experiments showing time activity recommendation is as given in C

**Loss Function Evaluation.** We examined the effect of different loss functions on link prediction, with results shown in Table 5. Performance is reported across Test ROC Accuracy, Test Precision, New Node ROC Accuracy, and New Node Precision. CTC Loss yields the best performance over-

Table 5: Comparison of loss functions on link prediction. CTC Loss achieves the strongest results across all metrics.

| Loss Function | Test ROC Accuracy | Test Precision | New Node ROC Accuracy | New Node Precision |
|---|---|---|---|---|
| Binary Cross Entropy | $0.6070 \pm 0.0134$ | $0.6947 \pm 0.0151$ | $0.6097 \pm 0.0192$ | $0.5722 \pm 0.0187$ |
| Cross Entropy Loss | $0.6163 \pm 0.0128$ | $0.7181 \pm 0.0143$ | $0.5808 \pm 0.0203$ | $0.6739 \pm 0.0198$ |
| Mean Squared Error | $0.7708 \pm 0.0102$ | $0.7575 \pm 0.0120$ | $0.4802 \pm 0.0210$ | $0.5282 \pm 0.0204$ |
| L1 Loss | $0.6173 \pm 0.0119$ | $0.6577 \pm 0.0154$ | $0.5660 \pm 0.0188$ | $0.6204 \pm 0.0196$ |
| Log Likelihood | $0.6561 \pm 0.0127$ | $0.6124 \pm 0.0149$ | $0.5909 \pm 0.0201$ | $0.5497 \pm 0.0189$ |
| **CTC Loss (Ours)** | **$0.8664 \pm 0.0061$** | **$0.8185 \pm 0.0079$** | **$0.7962 \pm 0.0094$** | **$0.8040 \pm 0.0102$** |

all, highlighting its strength in sequence alignment for temporally ordered prediction tasks. Cross Entropy performs competitively, especially in Test and New Node Precision, while Binary Cross Entropy and L1 provide moderate results. Mean Squared Error underperforms, confirming its limitations for categorical sequential data. Log Likelihood surpasses BCE but remains weaker than CE and CTC.

**Negative Sampling Strategies Comparison** We evaluated the impact of different negative sampling strategies on model performance for link prediction tasks. Negative sampling is crucial for distinguishing true connections from false ones, improving the model's ability to recommend future activities and maintain sequential memory structures. Table 6 presents results for three strategies: Random, Historical, and Inductive sampling. Random Sampling selects negative samples uniformly,

Table 6: Performance comparison of different negative sampling techniques for link prediction tasks. Metrics include Test ROC Accuracy, Test Precision, New Node ROC Accuracy, and New Node Precision.

| Negative Sampling Techniques | Test ROC Accuracy | Test Precision | New Node ROC Accuracy | New Node Precision |
|---|---|---|---|---|
| Random | $0.7725 \pm 0.0108$ | $0.7576 \pm 0.0124$ | $0.7335 \pm 0.0141$ | $0.7295 \pm 0.0150$ |
| Historical | $0.7650 \pm 0.0113$ | $0.7788 \pm 0.0130$ | $0.6990 \pm 0.0158$ | $0.7042 \pm 0.0164$ |
| **Inductive (Ours)** | **$0.8664 \pm 0.0061$** | **$0.8185 \pm 0.0079$** | **$0.7962 \pm 0.0094$** | **$0.8040 \pm 0.0102$** |

achieving moderate results (Test ROC: 0.7725) but lacks contextual awareness. Historical Sampling uses past data to improve precision (0.7788) but struggles with generalization. Inductive Sampling outperforms both, aligning samples with temporal context and achieving the best metrics (Test ROC: 0.80664), making it ideal for episodic memory and activity recommendations.

**Sampling Strategies** We analyzed the performance of different sampling techniques for link prediction tasks, including Uniform, Recent, and Time Interval Aware approaches (Table 7). Uniform sampling, which selects nodes uniformly, achieved moderate results (Test ROC: 0.7659) but failed to prioritize relevant temporal dynamics. Recent sampling, focusing on the latest interactions, strug-

Table 7: Comparison of sampling techniques for link prediction tasks, showing Test ROC Accuracy, Test Precision, New Node ROC Accuracy, and New Node Precision.

| Sampling Techniques | Test ROC Accuracy | Test Precision | New Node ROC Accuracy | New Node Precision |
|---|---|---|---|---|
| Uniform | $0.7659 \pm 0.0145$ | $0.7583 \pm 0.0123$ | $0.7739 \pm 0.0230$ | $0.7181 \pm 0.0099$ |
| Recent | $0.6750 \pm 0.0343$ | $0.7181 \pm 0.0412$ | $0.5808 \pm 0.0213$ | $0.6739 \pm 0.0013$ |
| **Time Interval aware** | **$0.8664 \pm 0.0061$** | **$0.8185 \pm 0.0079$** | **$0.7962 \pm 0.0094$** | **$0.8040 \pm 0.0102$** |

gled with generalization, particularly for unseen nodes (New Node ROC: 0.5808). The Time Interval Aware method outperformed others across all metrics (Test ROC: 0.80664, New Node ROC: 0.7962), as it effectively captured temporal patterns and contextual relevance, demonstrating its utility for sequential and time-sensitive tasks like activity anticipation and memory modeling.

## 6 CONCLUSION LIMITATIONS AND FUTURE WORK

The framework is relevant for prospective assistive agents designed to help users who may experience difficulty tracking or recalling daily activities, while also improving human–robot collaboration and task efficiency. By leveraging daily and yearly activity patterns, the model achieves state-of-the-art performance in predicting complex future action sequences. Key innovations include adaptive memory representations, advanced time encoding, and robust sampling strategies, enabling the system to dynamically utilize past experiences. These capabilities make the model applicable in real-world scenarios, such as assistive technologies, adaptive robotics, and personalized recommendations. The framework offers societal benefits by aiding individuals with memory impairments through reminders and navigation guidance, while improving human-robot collaboration and task efficiency. Limitations include dependency on recurring patterns and large datasets, which may hinder performance in sparse environments. Additionally, the computational complexity of multimodal data processing and dynamic memory updates poses challenges in resource-constrained settings. Future work will focus on optimizing for such scenarios, integrating reinforcement learning for context-aware decisions, and expanding the framework to incorporate additional modalities, such as physiological and environmental data.

## ACKNOWLEDGMENTS

The authors used a large language model (ChatGPT) solely to polish grammar and improve the clarity of writing. All research ideas, experiments, analyses, and conclusions are entirely the work of the authors.

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

## A    DETAILED DATASET CONSTRUCTION FROM EGO4D

Our long-horizon timeline dataset is constructed entirely from existing Ego4D annotations (Grauman et al., 2022; Zhou et al., 2025). Because Ego4D is composed of short, isolated egocentric clips rather than continuous lifelog recordings, it is essential to describe precisely how these clips are reorganized into a synthetic multi-year timeline. The goal of this procedure is to preserve annotation fidelity, avoid any form of information leakage, and ensure that no artificial temporal regularities are introduced.

**Source Videos and Filtering Criteria**    We begin by selecting all Ego4D clips that include both scenario-level and moment-level annotations as defined in `ego4d.json`. For each clip, we read the unique `video_uid`, duration, resolution, device metadata, as well as all scenario and moment annotations. The construction process strictly follows the official train/val/test partitions: no information from validation or test videos is ever used when building the training timeline.

**Scenario-Level Grouping**    Ego4D clips are annotated with high-level scenario categories such as cooking, construction tasks, or office work. These scenarios define the activity categories used in our timeline. When a single clip contains multiple scenario annotations, the temporally segmented structure provided by Ego4D annotators is retained exactly. We do not merge, reinterpret, or reassign scenarios at any point.

**Moment-Level Fine-Grained Actions**    Moment-level annotations capture fine-grained actions within each scenario. These are included verbatim in our dataset. Their timestamps, ordering, and boundaries are left unchanged. No smoothing, interpolation, or merging of actions is performed. This ensures that all temporal structure in the final timeline originates from Ego4D, not from synthetic post-processing.

**Activity-Centric Reorganization**    Our objective is long-horizon modeling of activities and actions rather than modeling the continuity of a specific identity or environment. For this reason, the identity of the camera wearer is never preserved across days, and no attempt is made to create visual or narrative continuity between clips. The timeline is therefore activity-focused: the same activity may occur many times, but almost always in a different environment, with different individuals, and under different visual contexts, reflecting the natural variability of Ego4D.

**Timetable-Style Organization of Activities**    To convert the set of disconnected Ego4D videos into a coherent three-year timeline, we rearrange clips into a timetable-like structure resembling everyday rhythms. Weekdays are associated with routine activities such as work, commuting, or household tasks, whereas weekends emphasize leisure, hobbies, social interactions, and less structured activities. For each synthetic day, we sample clips whose activity categories match the corresponding day type, and then place these clips in chronological order. The clips themselves are kept completely intact: their internal timing, order of events, and action boundaries remain exactly as provided by Ego4D. The reorganization thus produces a calendar-like sequence while preserving all original temporal information.

Table 8: Key differences between the original Ego4D dataset and our reorganized long-horizon timeline version.

| Aspect | Original Ego4D | Long-Horizon Timeline (Ours) |
|---|---|---|
| **Data Structure** | Short, independent egocentric clips with no temporal linking. | Clips reorganized into a 3-year timetable-style timeline. |
| **Annotations** | Scenario- and moment-level labels on isolated clips. | Same annotations preserved verbatim; no modification. |
| **Temporal Continuity** | No relations across clips. | Sequential placement into synthetic days; clip boundaries unchanged. |
| **Identity / Environment** | Identity/environment consistent only within each clip. | No cross-day identity continuity; activity-centric timeline. |
| **Split Integrity** | Uses official train/val/test partitions. | Strictly respects splits; no cross-split usage. |
| **Scenario Balance** | Natural Ego4D frequency (e.g., cooking-heavy). | Mild scenario balancing during sampling; videos unchanged. |
| **Storage Format** | Individual clips with metadata. | One directory per synthetic day + `day_meta.json`. |

**Preventing Statistical Artifacts**  A number of safeguards are applied to ensure that the timeline does not inadvertently contain artifacts that a model could exploit.Additional minutes are needed to complete a day, new clips are sampled rather than reusing existing ones. Hard boundaries between clips are maintained without any blending. If a particular scenario category is exhausted within the constraints of a split, the corresponding day simply contains fewer minutes of activity rather than being padded with recycled content. These choices prevent the formation of trivial repetitions or artificial long-term regularities.

**Dataset Size, Duration, and Balancing**  The final timeline covers 1,095 days. The duration of each day varies depending on the lengths of the sampled clips, naturally producing uneven distributions across the calendar. To avoid dominance by highly frequent Ego4D scenarios such as cooking or hand-manipulation tasks, we introduce mild balancing by adjusting the sampling probabilities of scenarios. Importantly, no video is modified or rescaled during this process; only the sampling frequency of scenarios is altered.

**Storage Format and Reproducibility**  Each synthetic day is represented by a directory containing a `day_meta.json` file detailing the selected videos, their ordering, and the exact timestamps at which they appear in the timeline. Scenario-level and moment-level annotations are included alongside pointers to the original Ego4D videos, which are not duplicated. Every component of the dataset is fully traceable back to its source in Ego4D, ensuring maximal transparency and reproducibility.

**Summary of Safeguards**  In summary, the timeline uses only original Ego4D content; no synthetic data are generated. The reorganization preserves all annotations and focuses on activity-level structure rather than identity or environment continuity. Clips may share activity categories but differ in visual context and camera wearer. Through these constraints, the resulting multi-year timeline provides a controlled yet realistic setting for long-horizon episodic-memory modeling.

## B  ILLUSTRATIVE SCENARIO

We illustrate our system through a simplified three-day sequence in which the agent observes and supports the user ("master") across daily activities. In addition to fine-grained actions, the system also monitors *coarse, timestamp-like high-level activities* (e.g., `04:00--sleeping`, `08:00--morning_routine`, `13:00--lunch`), which serve as temporal anchors for long-horizon reasoning. An overview of the multimodal inputs and the agent's temporal-graph outputs is shown in Fig. 4.

**Day 1:** At `08:00`, corresponding to the master's `morning_routine` block, the master enters the kitchen. The agent tracks fine-grained actions such as opening the fridge, pouring milk, stirring cereal, and drinking coffee. By `08:30`, the master transitions into the `commute/work` block and leaves for the office. Coarse events like `13:00--lunch` and `20:00--guitar_practice` are also recorded.

**Day 2:** High-level timestamps follow a similar rhythm: `08:00--morning_routine`, `13:00--lunch`, `20:00--guitar_practice`. Within these blocks, minor variations (e.g., pouring milk at `08:01` or playing guitar slightly later) help the agent learn routine stability while

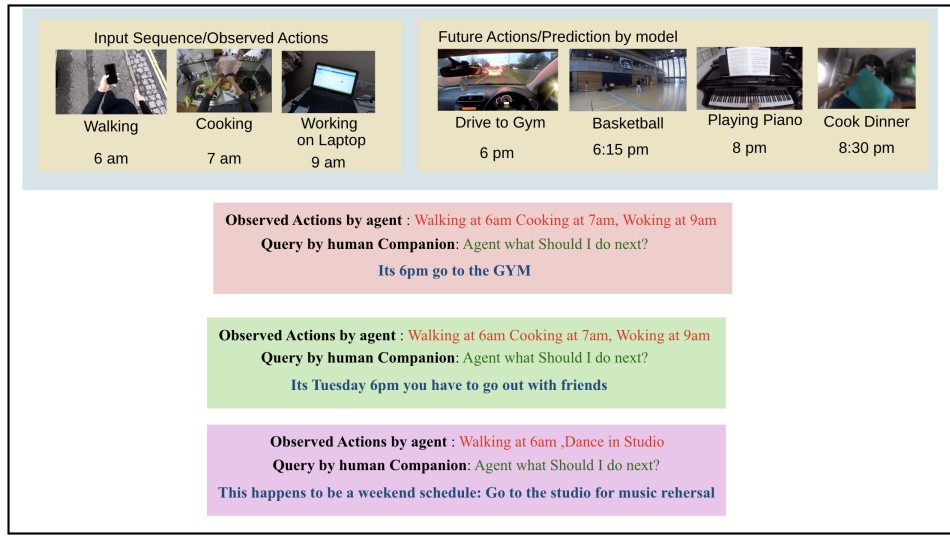

Figure 3: Overview of our illustrative scenario: the agent receives multimodal egocentric inputs (left) and produces temporal-graph memory representations and predictions (right).

**Input actions:**
06:00 | walking | outside,
07:00 | cooking | kitchen,
10:00 | art work | house,
13:00 | music | room
**Model predicts next actions:**
→ 13:05 | sewing | --

**Input actions:**
06:00 | walking | outside,
07:00 | cooking | kitchen,
10:00 | workshop work | workshop,
13:00 | coffee making | kitchen
**Model predicts next actions:**
→ 13:05 | playing games | --

**Input actions:**
06:00 | walking | outside,
07:00 | cooking | kitchen,
10:00 | cooking | kitchen,
13:00 | music | house
**Model predicts next actions:**
→ 13:05 | office work | --

**Input actions:**
06:00 | walking | outside,
07:00 | washing utensils | kitchen,
10:00 | music | room,
13:00 | coffee making | kitchen
**Model predicts next actions:**
→ 13:05 | painting | --

**Input actions:**
06:00 | walking | outside,
07:00 | washing utensils | kitchen,
10:00 | music | room,
13:00 | coffee making | kitchen
**Model predicts next actions:**
→ 13:05 | eating food | --

**Input actions:**
06:00 | walking | outside,
07:00 | using phone | house,
10:00 | kitchen activity | kitchen,
13:00 | music | room
**Model predicts next actions:**
→ 13:05 | painting | --

**Input actions:**
06:00 | cooking | kitchen,
07:17 | shopping | supermarket,
08:54 | coffee making | kitchen,
10:45 | music | house
**Model predicts next actions:**
→ 10:50 | eating food | --

**Input actions:**
06:00 | walking | outside,
07:17 | cutting vegetables | room,
08:54 | coffee making | kitchen,
10:45 | music | room
**Model predicts next actions:**
→ 10:50 | sewing | --

Figure 4: Example of the model's activity recommendation output, where the agent suggests probable next actions using its learned long-horizon temporal memory.

accommodating natural variability.

**Day 3:** Early morning (`04:00--sleeping`) and breakfast (`08:00--morning_routine`) follow familiar patterns. However, instead of leaving at `08:30`, the master begins remote work in the `09:00--focused_work` block. New events appear, such as a `17:00--walking` period and an earlier dinner. The agent adaptively interprets these timestamp-level segments to adjust expectations (e.g., recognizing that the user remains in a work block).

The agent encodes these episodes as temporal graphs built from multimodal streams (audio, visual, speech). Fine-grained actions form nodes (e.g., `pour_milk@08:01`), while high-level blocks

1. Cooking Baking
**Input actions:**
clean flour, put cutter, put pot, take container, put container, close cabinet, move pot, remove tray
**Model predicts next actions:**
→ preheat oven, mix batter, pour batter

2. Table-Setting Example
**Input actions:**
clean floor, adjust table, put table, pull bucket, dip bucket, squeeze bucket, unfold napkin, clean napkin
**Model predicts next actions:**
→ place plates, put forks, put cups

3. Construction / Repair Example
**Input actions:**
cut soil, mold cement, fill cement, scrape cement, put cement, adjust brick, move brick, turn brick
**Model predicts next actions:**
→ align bricks, apply more cement, smooth surface

4. Gardening Example
**Input actions:**
put plant, put plant, take plant, cut plant, put plant, take trowel, scoop bag, pour soil
**Model predicts next actions:**
→ press soil, water plant, place pot in sun

5. Soldering Example
**Input actions:**
take solder-iron, put solder-iron, hold wire, take solder-iron, attach wire, attach wire, put solder-iron, take rubber-band
**Model predicts next actions:**
→ heat joint, apply solder, cool connection

6. Ironing Clothes Example
**Input actions:**
put iron, adjust shirt, take iron, press shirt, put iron, adjust shirt, take shirt, put cloth
**Model predicts next actions:**
→ fold shirt, hang shirt, store iron

Figure 5: Example of the model's action recommendation output, where the agent suggests probable next actions using its learned long-horizon temporal memory.

provide contextual nodes (e.g., `morning_routine@08:00`). Timestamped edges capture both sequential and hierarchical structure. As new days unfold, message passing updates graph states while preserving long-horizon temporal coherence.

Such representations enable:

**Next-Action Anticipation:** Predicting likely next steps by combining fine-grained and high-level temporal cues (e.g., recognizing that after a `04:00--sleeping` block, the next event at `08:00` is breakfast).

**Memory Support:** Offering prompts when the user hesitates (e.g., *"It's 08:00; your morning routine usually begins with cereal—proceed?"*).

**Procedure Recovery:** Retrieving observed procedural sequences to answer queries such as *"How do I make tea?"*.

**Context-Aware Suggestions:** Leveraging high-level timestamps to provide timely support (e.g., *"It's almost 20:00—ready for guitar practice?"* or *"You often walk around 17:00."*). The model's predicted next activities are illustrated in Fig. 4.The model's predicted next actions are illustrated in Fig. 5.

## C ANALYSIS OF TIME–PLACE SENSITIVITY IN INTERACTION PATTERNS

To examine what temporal and spatial regularities the model captures during training, we analyze how activities co-occur with both the *time of day* and the *location* in which they are performed. Human activity sequences naturally exhibit strong dependencies on these factors—for example, visiting the kitchen at 7 AM is typically associated with preparing breakfast, while entering the garden in the late afternoon often corresponds to watering plants or engaging in light outdoor work. If the model learns such patterns, it demonstrates an ability to reason over higher-level contextual structure rather than relying solely on local frame-based cues.

**Co-occurrence heatmaps.** Using 16 days of interaction data, we compute activity–hour co-occurrence matrices for both ground-truth labels and model predictions, and visualize them as heatmaps. These maps reveal clear temporal routines in the human's behavior (e.g., *cooking* peaks around morning and early evening, *office work* is concentrated between 10–16 hours, and *hobbies* occur predominantly in evening hours), and the model reproduces many of these regularities.

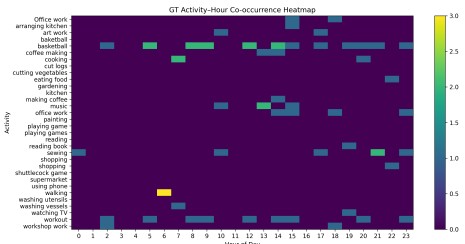 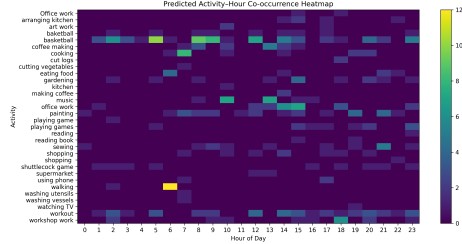

(a) Ground-truth activity–hour distribution.      (b) Model-predicted activity–hour distribution.

Figure 6: Comparison of (a) ground-truth and (b) model-predicted activity–hour distributions.

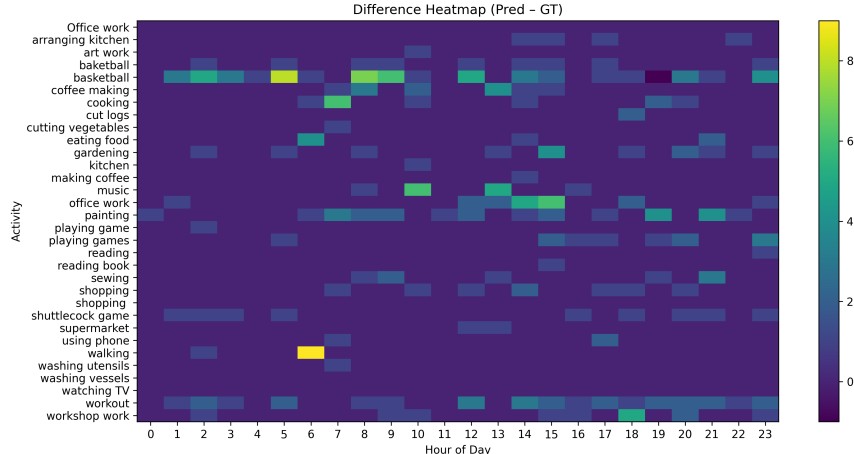

Figure 7: Difference heatmap (Predicted – Ground-truth). Red regions indicate overestimation of activity occurrences at specific hours, whereas blue regions indicate underestimation.

**Temporal and spatial sensitivity.** Across activities, the model demonstrates strong sensitivity to the interplay between time and place. Temporal patterns alone cannot fully disambiguate activity, but when combined with location cues, they become highly predictive. For example:

- entering the **kitchen at 07:00** strongly corresponds to *cooking or making breakfast*;
- being in the **work room at 11:00** aligns with *office work*;
- being **outside around 17:00** often precedes *gardening*;
- being in the **living room at 21:00** typically precedes *relaxation activities* such as reading or watching television.

**Interpretation.** These patterns suggest that the model has learned not only short-term transitions between actions but also higher-level routines grounded in both temporal and spatial context. This is crucial for long-term anticipation: the model can infer that after observing a person enter the kitchen in the early morning, the probability of a cooking-related action is high, whereas the same location visited at 14:00 may indicate cleaning or preparing a snack. Thus, the model leverages time and place jointly to structure its predictions, capturing meaningful regularities in human daily routines.

