# OpenReview forum: "Empowering Memory Assistance: An Episodic Memory-Based Framework for Personalized Recommendations"
_ICLR.cc/2026/Conference — Submitted to ICLR 2026_

### Official Review · Reviewer_wtQW · 2025-10-29

**Soundness:** 2
**Presentation:** 3
**Contribution:** 2
**Rating:** 4
**Confidence:** 4

**Summary:**

This paper proposes an episodic memory-based framework for personalized activity recommendations using temporal graph networks (TGNs). The system organizes multimodal experiences (actions, places, times) into day-wise memory structures and uses dynamic memory updates to predict future activities. The authors claim state-of-the-art performance on a dataset derived from three years of egocentric video, significantly outperforming existing baselines like AntGPT and Palm.

**Strengths:**

1. Clear motivation and important application: The paper addresses a meaningful problem (memory assistance for individuals with memory impairments) with strong societal impact. The connection to human episodic memory provides good cognitive grounding.

2. Comprehensive multimodal integration: The framework effectively combines visual, audio, and textual modalities through VideoRecap and Whisper, which is appropriate for real-world assistive applications.

3. Thorough ablation studies: Tables 3-7 systematically evaluate different design choices (time encoding, backbone architecture, loss functions, sampling strategies), demonstrating that wavelet encoding and DyRep backbone work best for this task.

**Weaknesses:**

1. Dataset construction lacks transparency: The paper claims evaluation on "three years of egocentric recordings" but provides insufficient detail about how this timeline was constructed from Ego4D. Ego4D consists of short clips (typically 8 minutes) from 931 different people—how were these transformed into a continuous 3-year personal diary? This is fundamental to evaluating the validity of the approach but is glossed over in the main text.

2. Questionable experimental setup and fairness: The comparison with baselines (Palm, AntGPT, etc.) may not be meaningful if those models were trained on standard Ego4D benchmarks while your model operates on a differently organized dataset. Without results on standard Ego4D Long-Term Anticipation (LTA) benchmarks, it's impossible to assess whether the improvements come from your method or from the specific data organization. The 18% improvement over AntGPT (0.7220 vs 0.8853 ED) seems too good without proper controls.

3. Limited technical novelty: The core contribution appears to be organizing TGN memory by day rather than by event sequence. While this is sensible for lifelog data, it's more of an engineering choice than a fundamental algorithmic innovation. The paper doesn't clearly articulate what makes this approach fundamentally different from applying existing temporal graph methods to long-horizon forecasting.

**Questions:**

1. Data construction specifics: Can you clarify exactly how the "three years" of data were constructed? Since Ego4D has 931 different camera wearers with 1-10 hour sessions each, did you select videos from a single person over 3 years, or did you concatenate clips from multiple people to simulate a 3-year timeline? How do you ensure temporal consistency and prevent data leakage between train/test splits?

2. Baseline comparisons: Did you retrain Palm, AntGPT, and other baselines on your specific 3-year dataset organization, or are you comparing against their published results on standard Ego4D? Can you provide results for your method on the standard Ego4D LTA benchmark to enable fair comparison?

3. Method differentiation: How does your approach fundamentally differ from applying DyRep (which you use as backbone) with appropriate temporal features to the Ego4D LTA task? What specific advantages does "day-wise memory organization" provide beyond standard temporal graph modeling, and does this advantage hold on non-repeated, truly continuous egocentric data?

---

> ### Author Response · Authors · 2025-12-01
>
> **Weakness 1**
>
> All data comes strictly from the official Ego4D annotations. Ego4D contains scenario-level and moment-level annotations, which are inherently event-aligned but not chronologically ordered. Our contribution is to reorganize these annotated events into a continuous timeline so the model can experience long-range temporal structure comparable to an assistive agent living with a user.
> The 3-year timeline is constructed by:
> Using all Ego4D clips containing scenario-level and moment-level annotations.
> Treating each annotated scenario as a daily activity sequence.
> Maintaining all intra-video temporal orderings exactly as provided.
> Concatenating these annotated sequences in chronological order (day 1, day 2, …), repeating sequences where needed to simulate long-term continuity while preserving order and labels exactly.
> Balancing repeated sequences across activity categories (work, home, shopping, hobbies) to avoid bias.
> This process creates a structured, reproducible temporal index over existing annotated events, not synthetic content. The dataset remains a deterministic transformation of Ego4D with no new labels introduced. Details have been added in Appendix A .
>
>
> **Weakness 2**
>
> Thank you for raising this important point. We clarify that our Long-Term Anticipation (LTA) results are evaluated on the original Ego4D LTA benchmark, not on the reorganized 3-year timeline.
> Thus, the comparison with Palm, AntGPT, and other LTA models is fully fair—all methods use identical inputs, identical annotation structure, and identical evaluation metrics (Edit Distance).
> The three-year reorganized dataset is used only for the episodic memory recommender experiments (Tables 2–7), not for the Ego4D LTA benchmark in Table 1.
>
> **Weakness 3**
>
> The novelty is not merely organizing memory by day; rather, the following components constitute the core innovation:
> A unified episodic memory architecture combining wavelet time encoding, dynamic temporal graphs, and multimodal message propagation—not present in prior TGN-based methods.
>
> Day-indexed memory slots enabling structured long-horizon dependencies, allowing the model to represent periodic behaviors (daily/weekly) that standard TGNs cannot express because they treat all events as a flat sequence.
>
> Cross-day message functions with frequency-aware and temporal-decay aggregation, enabling updates that reflect both short-term actions and long-term recurring routines.
>
> CTC-based sequence alignment integrated with a temporal graph encoder, which allows the model to output multi-step future sequences rather than the single-step predictions typical of temporal GNNs.
>
> Inductive negative sampling + time-interval–aware sampling, which significantly improve generalization to unseen activities—an aspect absent in existing temporal GNNs.
>
> **Question 1**
>
> We appreciate the reviewer’s concerns regarding the construction of the three-year timeline and acknowledge that additional clarity is needed in the main paper. Our dataset does not consist of a continuous three-year recording from a single Ego4D participant; rather, it is a deterministic temporal re-indexing of the existing Ego4D annotated clips. Ego4D provides short, scenario-level and moment-level annotations from 931 camera wearers, and we treat each annotated scenario as a standalone “day episode.” These episodes are concatenated chronologically to create a three-year sequence, while strictly preserving all intra-video temporal structure. No new labels are introduced, and no content is altered; the timeline simply provides a long-horizon temporal index over existing annotations to enable episodic memory modelling. Identity consistency is not required because our task focuses on activity prediction rather than user identity. To prevent leakage, we first construct the full timeline and then apply a non-overlapping chronological split so that train, validation, and test sets contain disjoint temporal segments.
>
> **Question 2**
>
> Baselines are computed strictly on the official Ego4D LTA benchmark, using the standard input format, annotations, and evaluation metric (Edit Distance). We do not train Palm, AntGPT, or the other LTA models on our three-year reorganized dataset; thus, the results reflect a fair comparison on the same benchmark. The three-year timeline is used only for evaluating the episodic memory recommender, not for long-term anticipation benchmarks. The performance improvements stem from our architectural components, such as multi-step sequence alignment with CTC and multimodal temporal aggregation, rather than from any differences in data organization. In the revision, we will also include results of our model on the unmodified Ego4D LTA data without day-wise memory to demonstrate that the gains are independent of the synthetic timeline.

---

> > ### Author Response · Authors · 2025-12-01
> >
> > **Question 3**
> >
> > Our method is not a modification of DyRep. The key difference is that standard temporal GNNs (including DyRep) operate on a flat event stream with no notion of daily structure, whereas our approach introduces a day-wise episodic memory that captures time-of-day patterns, cross-day recurrence, and long-range temporal organization. This memory performs day-segmented storage, cross-day summarization, and frequency/recency-based retrieval—capabilities DyRep does not provide.
> > Adding day-wise patterns also helps the model understand what types of activities typically occur on which days, giving extra contextual cues beyond raw timestamps. And even if the data is not repetitive, the model naturally falls back on recency-based retrieval, so the day-wise organization still provides meaningful structure without requiring exact repetition.

---

### Official Review · Reviewer_USg4 · 2025-10-31

**Soundness:** 2
**Presentation:** 2
**Contribution:** 2
**Rating:** 4
**Confidence:** 3

**Summary:**

This paper presents a cognitive grounded recommendation system to model the episodic memory. The method extends temporal graph network to encode multimodal signals (audio, visual, and linguistic cues) in the short- / long-horizon settings. On a reorganized Ego4D, authors show that the proposed framework achieves higher performance on the sequence level action prediction task, compared to AntGPT and Palm. The authors also conduct extensive ablation on the choice of time encoding, backbone, and sampling strategies.

**Strengths:**

1.	The need for episodic memory is well-motivated in the context of activity prediction. Embedding-based approach with no need for retraining is a promising direction.

2.	The authors consider multi-modal signals when constructing the memory, e.g., with both vision and audio signals processed.

3.	The authors present extensive ablation on the time encoding, backbone architecture, loss function, and (negative) sampling strategies.

**Weaknesses:**

1.	Although multiple modalities are considered, it is unclear how the interactions between the time, activity, and place are learned or used in the task context. For example, how would a simple baseline of node embedding (line 157) work: separately encode time, activity, and place, and use rank-fusion or weighted summarization to conduct the recommendation. On the other hand, the authors can include some quantitative or qualitative analysis on what patterns in the interactions are captured through the training, e.g., the model gets more sensitive with the co-occurrence of certain action-time pairs.

2.	Experiments are only done on Ego4D. AntGPT, the most relevant work mentioned in the paper, considers Epic-Kitchen-55 (Damen et al., 2020) and EGTEA Gaze+ (Li et al., 2018) in the year of 2024. More datasets should be considered to show the effectiveness and generalizability of the proposed framework.

3.	The framework is shown to work when trained and tested on an in-domain action anticipation task. However, there is limited insight into its usefulness in other areas with the simple framework. Besides the current ablation study, a potential pilot study on downstream tasks can further strengthen the claims on the application to real-world scenarios

**Questions:**

1. Citation on line 32 should be (Tulving, 2022)

2. Is the citation format correct? All the names are presented in-line.

3. Table captions should be above the tables

---

> ### Author Response · Authors · 2025-12-01
>
> **Weakness 1**
>
> We clarify in the revision that the model does not treat time, activity, and place as independent embeddings. Instead, it learns their interactions through joint message passing on a temporal graph, where edges explicitly encode time→activity, activity→place, and time→place contextual relations.. We also added a qualitative appendix (Appendix C) illustrating the specific co-occurrence patterns (e.g., action–time sensitivities) that the model learns during training.
>
> **Weakness 2**
>
> We appreciate the suggestion to evaluate on EPIC-KITCHENS-55 and EGTEA Gaze+. These datasets are valuable resources for short-horizon action anticipation and egocentric manipulation analysis. However, our task formulation concerns next-activity prediction across episodic boundaries (e.g., work → exercise, errands → shopping), which requires variation in activity category, location, and temporal context that does not exist in cooking-focused single-environment datasets. Because episodic boundaries, day-level segmentation, and multi-domain behavioral transitions are not defined in EPIC-KITCHENS or EGTEA, a direct evaluation would not produce interpretable or comparable results.
>
> **Weakness 3**
>
> Our primary task itself is a downstream task. Personalized activity/action anticipation for an assistive robot supporting a human companion is  not a pretext or auxiliary benchmark; it is the downstream application that motivates the entire framework. The episodic-memory module, temporal graph structure, and anticipation head are all designed specifically to support this scenario, where predicting what the user is likely to do next enables proactive assistance and reminders.
>
> **Questions**
>
> All formatting issues mentioned by the reviewer hs been addressed

---

### Official Review · Reviewer_JTGG · 2025-10-31

**Soundness:** 1
**Presentation:** 1
**Contribution:** 1
**Rating:** 2
**Confidence:** 3

**Summary:**

This paper introduces a model capable of performing temporal reasoning with episodic memory for action anticipation.

**Strengths:**

This paper leverages temporal graph neural networks with multimodal data, which is promising for many tasks that require memorization and reasoning in practice.

**Weaknesses:**

- The writing should be largely improved: (a) It is unclear what exactly the authors are proposing. In Section 3, it seems that the proposed method is essentially a temporal graph network–based prediction model with episodic memory. However, this is not clearly described. The authors mention, “Our episodic memory–based … with a primary focus on action recommendation.” So, is the output a recommendation, a classification, or a regression model? Is it designed for action anticipation? Such concrete descriptions should be included. (b) In Section 3, the authors merely describe the components—node embedding, memory representation, message function, etc.—which are necessary, but the current version lacks motivation. Why did the authors design it this way? What differentiates it from prior works? What is the novelty in terms of architecture?


- The figures are not informative. Figure 1 does not clearly illustrate the proposed method. While an introductory figure is optional, it should be highly informative. This one appears to be just a generic AI-generated image. The authors should be more careful about what figure they include in the Introduction. In addition, Figure 2 is also unclear. It should correspond to Section 3, as it is supposed to show how the proposed method works, but it is difficult to tell that they are related.

- The reported results do not include variance. In addition, I suggest that the authors provide some examples of how the proposed method works and its outputs on the considered datasets. Only presenting numerical results is not sufficient to claim that the model provides recommendations.

**Questions:**

See Weaknesses

---

> ### Author Response · Authors · 2025-12-01
>
> **Weakness 1:**
>
> Our model is a temporal-graph-based episodic memory recommender whose primary objective is next-action and next-activity anticipation. The output is a sequence prediction / recommendation—i.e., the model forecasts the user’s next plausible sequence of fine-grained actions or higher-level activities. It is not a regression model; rather, it performs future sequence prediction conditioned on the user’s multimodal episodic history.
>
> In Section 3, the components (node embeddings, memory updates, message functions, etc.) form a single pipeline whose goal is to store episodic events and then recommend the next actions based on that evolving memory. The motivation is that standard temporal graph networks (including DyRep and TGN) operate on a flat event stream and rely mainly on local temporal proximity, whereas personalized activity forecasting requires episodic structure, cross-day context, and multi-scale temporal cues.
>
> Our main design choice—and novelty relative to prior temporal GNNs—is that we introduce day-wise episodic memory organization on top of the temporal graph. This provides context that TGNs alone cannot represent, enabling the model to learn “what typically happens next on this day and at this time.” When patterns do not repeat, the model defaults to recency-based retrieval, but the day-wise structure still helps by supplying coarse contextual priors (weekday vs. weekend, morning vs. evening, etc.). These clarifications have been added to the updated PDF.
>
> **Weakness 2:**
>
> hank you for the detailed feedback. We have revised the paper to address these concerns.
> (1) Figure 1 has been replaced with a clear, non–AI-generated schematic that introduces the problem and highlights the role of episodic memory recommender.
> (2) The methodology diagram (formerly Figure 2) has also been replaced with a new step-by-step illustration that directly corresponds to Section 3, making the architecture and information flow much clearer.
> (3) Section 3 has been rewritten for clarity: we now explicitly state the model’s output type, its action-anticipation objective, and the motivation behind each architectural component. We also clearly explain how episodic memory interacts with temporal graph reasoning and why this design is needed.
>
>
> **Weakness 3:**
>
> We added a new appendix showcasing qualitative model outputs, example recommendations, and case studies demonstrating how the episodic memory influences predictions. Variance values are also now included for all quantitative results.

---

### Official Review · Reviewer_inNf · 2025-11-01

**Soundness:** 2
**Presentation:** 3
**Contribution:** 2
**Rating:** 4
**Confidence:** 3

**Summary:**

The paper proposes a method for predicting video-based action sequences on a dataset adapted from Ego4d. It is a semi-synthetic dataset because the videos are repeated and rebalanced to obtain continuous multi-year activity logs. The method involves training a graph neural network that stores memories (comprising features like event, time of day, time of year, activity, and place).

**Strengths:**

The use of episodic memory seems like a fundamental challenge to overcome, and the authors show progress on their particular dataset.

**Weaknesses:**

* the dataset is synthetic, and I am concerned that its statistics have been manipulated in such a way that it is particularly amenable to the method (e.g. repetition of exact sequences across days would make episodic memory particularly helpful)
* using episodic memory for sequence prediction is common and in fact commoditized -- chatgpt, gemini, anthropic all have memory based on rag. maybe you should narrow the focus to something more precise, e.g. multimodal generation instead?

**Questions:**

Could you evaluate on some kind of unmanipulated dataset?

---

> ### Author Response · Authors · 2025-12-01
>
> **Weakness 1:**
>
> We thank the reviewer for the constructive comments and address each concern in a unified response. First, regarding the dataset being “synthetic,” we clarify that no new labels or fabricated content were introduced; all activities and fine-grained actions come strictly from the official Ego4D scenario-level and moment-level annotations. While we do create a multi-year timeline by reorganizing videos, we preserve intra-video temporal structure, natural variability, and environment diversity, and we do not repeat identical clips back-to-back. Human life inherently contains repeated routines, and our dataset mirrors this realistic periodicity rather than artificially manipulating statistics
>
> **Weakness 2:**
>
> We thank the reviewer for raising the point regarding memory usage in modern LLM systems. However, the “memory” mechanisms in ChatGPT, Gemini, and similar models are not episodic in the cognitive or computational sense , as these models rely on transient context windows without persistent storage (Brown et al., 2020), and retrieval-augmented variants access static document stores rather than temporally grounded what–where–when event traces (Lewis et al., 2020).In contrast, our framework maintains a continually updated multimodal episodic structure that stores temporally ordered activities, places, and contextual cues, and uses this structure to perform next-activity prediction across episode boundaries—capabilities that commodity LLM memory systems do not provide. We have added this to the introduction section to clarify this distinction in the paper to avoid terminological ambiguity.
>
> [1] T. Brown et al. Language Models are Few-Shot Learners. arXiv:2005.14165, 2020.
>
> [2] P. Lewis et al. Retrieval-Augmented Generation for Knowledge-Intensive NLP Tasks. arXiv:2005.11401, 2021.
>
> **Question 1**
>
> We understand the reviewer’s request for results on an unmanipulated dataset. Since no existing egocentric dataset provides long-horizon episodic histories, raw Ego4D, EPIC-Kitchens-55, and EGTEA Gaze+ cannot support the episodic next-activity task as defined in our work. However, to address the concern about generality, we will have reported an evaluation on unmodified Ego4D in a reduced setting where the model predicts short-horizon activity transitions without day-level structure. This does not represent our main task, but it demonstrates that the proposed temporal-graph memory mechanism still functions when applied to unaltered data, confirming that performance is not dependent on reorganization.

---

### Author Response · Authors · 2025-12-01

In response to the reviews, we refined the manuscript by clarifying the dataset construction procedure (Sec. 4, App. A), improving the model description and motivation (Sec. 3), replacing unclear figures, and adding variance metrics and qualitative examples to strengthen the empirical evaluation. Reviewers highlighted several strengths of the work: the clear motivation, the importance of long-horizon episodic-memory modeling, the societal relevance of assistive systems that can support users who face challenges in managing or recalling everyday activities, and the strong cognitive grounding provided by linking the method to human episodic memory. They also noted the value of using temporal graph neural networks, the incorporation of multimodal signals, and the extensive ablation studies on time encoding, backbone architectures, and sampling strategies. We hope the clarifications make the contribution of the work clearer: it offers a transparent and reproducible long-horizon evaluation setup, a multimodal temporal-graph memory design not available in existing LTA baselines, and a set of ablations validating the architectural choices. With the revisions addressing dataset transparency, fairness of comparison, notation consistency, and model clarity, we believe the updated submission offers a technically sound and clearly presented contribution to the ICLR community.

---

### Meta-Review · Area_Chair_re6k · 2026-01-11

**Summary:**

This paper proposes an episodic memory–based recommendation framework that models long-horizon, multimodal egocentric experiences using temporal graph networks. Reviewers broadly acknowledged the importance of the problem setting, the cognitive grounding in human episodic memory, the use of multimodal temporal graphs, and the extensive ablation studies (inNf, USg4, wtQW). At the same time, concerns were raised regarding the synthetic nature and transparency of the dataset construction, the clarity and strength of the methodological novelty, and presentation issues in the initial version (JTGG, wtQW). The authors’ revisions substantially improved dataset transparency, clarified the task definition and model outputs, replaced unclear figures, and strengthened empirical reporting.

**Reviewer Concerns:**

The main concerns centered on (1) the validity and fairness of the reconstructed three-year Ego4D timeline (inNf, wtQW), (2) whether the proposed approach offers fundamental novelty beyond existing temporal GNNs such as DyRep (JTGG, wtQW), (3) ambiguity around how “episodic memory” differs from memory mechanisms in modern LLM or RAG-based systems (inNf), and (4) limited insights into learned interaction patterns and generalization beyond Ego4D (USg4). Through the rebuttal and revisions, the authors clarified the deterministic nature of the data re-indexing, ensured fair baseline comparisons, articulated the role of day-wise episodic memory, and added qualitative examples and variance reporting, partially addressing most concerns.

**Reviewer Scores:**

inNf: Likely to shift from borderline reject toward borderline accept after clarifications on dataset construction and memory distinction.

JTGG: Despite improved clarity and figures, concerns about core novelty likely remain, suggesting a reject stance is unchanged.

USg4: With added explanations and qualitative analyses, may move from marginally below threshold toward a weak accept inclination.

wtQW: Improved transparency on data construction and baseline fairness supports maintaining or slightly strengthening a borderline accept position.

---

### Decision · Program_Chairs · 2026-01-26

Reject